# Racial disparities in cardiac transplantation: Chronological perspective and outcomes

**Jaimin R. Trivedi** [ID]*, Siddharth V. Pahwa, Katherine R. Whitehouse, Bradley M. Ceremuga, Mark S. Slaughter [ID]

Department of Cardiovascular and Thoracic Surgery, University of Louisville School of Medicine, Louisville, KY, United States of America

* Jaimin.trivedi@louisville.edu

## Abstract

### Background

The objective of this study was to evaluate annual heart transplant volumes and 3-year post-transplant outcomes since establishment of United Network for Organ Sharing (UNOS) database stratified by race.

### Methods

The UNOS thoracic transplant database was evaluated for adult patients since 1987. The available database was then stratified by Race: Black, White and Other and era of transplant: group 1(1987–1991), group 2(1992–1996), group 3(1997–2001), group 4(2002–2006), group 5(2007–2011), group 6(2012–2016) and group 7(2017 and later). Demographic and clinical factors were evaluated.

### Results

A total of 105,266 adults have been listed since 1987 and 67,824 have been transplanted. Of the transplanted patients 11,235 were Black, 48,786 White and 6803 were of Other race. The proportion of Black patients listed increased from 7% in 1987 to 13.4% in 1999 and 25% in 2019 and those transplanted increased from 5% in 1987 to 13.4% in 2001 and 26% in 2019. The survival of Black patients gradually improved.

### Conclusion

Historically, fewer Black patients received cardiac transplantation however, their access gradually improved over the years and account for over 25% of cardiac transplantations performed in recent years. The historically poor survival of Black patients has recently improved and became comparable to the rest.

**Data Availability Statement:** The data was obtained under a DUA from the HHS/AHRQ and is not permitted to share without prior authorization with any other entity. The authors have included

this in the methods sections. The data can be requested at the following web address "https://optn.transplant.hrsa.gov/data/request-data/". The following link provides data request instructions and fees. " https://optn.transplant.hrsa.gov/data/request-data/data-request-instructions/".

**Funding:** The authors received no specific funding for this work.

**Competing interests:** The authors have declared that no competing interests exist.

## Introduction

Since the 1st heart transplant was performed over a half century ago it has become the gold standard therapy to treat end-stage refractory heart failure. The field of heart transplantation has evolved significantly from the introduction of cyclosporine to improved donor organ preservation to use of mechanical circulatory support (MCS) devices and now ex-vivo perfusion of the donor heart [1–6]. All these incremental changes have improved waiting-list as well as post-transplant survival and expanded indications for recipient listing, and increased donor organ utilization, as over 3000 heart transplants are now performed annually in the United States since 2016 [5, 7]. Still, the ever-increasing burden of heart failure (over 6 million patients) has kept donor organ shortage high [8].

Few reports in the past have evaluated racial disparities in heart transplants, and those that have primarily focused on early post-transplant outcomes stratified by race [9, 10]. These studies have suggested adverse post-transplant outcomes for the African American population with or without the use of MCS and mostly using data of patients prior to 2010. There are a limited number of studies providing chronological perspective of racial disparities in access to heart transplant and surgical outcomes. The objective of this study was to evaluate annual heart transplant volume since the establishment of United Network for Organ Sharing (UNOS) database stratified by race. This study also aims to identify 3-year post-transplant outcomes of the patients stratified by race and period of transplant.

## Methods

### Data and study population

The UNOS thoracic transplant database was established in 1987 and has been continuously maintained since, with the addition of new variables over time. We used data from 1987 through June 2020 and queried for patients aged 18 years or older who were listed for heart transplant [11]. The University of Louisville IRB approved our study as an Exempt and the requirement for consent was waived. The UNOS has a data request portal through which a transplant center can request the data. We used this portal and requested the data which was available as a de-identified dataset. A data use agreement was signed with the UNOS to maintain the data integrity. The UNOS registry maintains its own data verification and validation tools to audit and keep the integrity of the data. There were less than 1% missing data points in terms of demographic, etiology, risk factors and outcomes. Factors such as use of MCS devices and some of the hemodynamic data were collected from 1990s onwards hence there were 10% missing data points in terms of MCS use and hemodynamic data. We did not impute missing data and excluded the patients from analysis. There have been 192 heart transplant centers since 1987 (some of them are now defunct) and have listed 105,266 patients. We excluded patients who were listed for heart-lung transplant. The available database was then classified in 3 groups based on the race of the listed patients: Black, White and Other. We also stratified the data in 7 chronological groups based on the period of transplant: group 1 (1987–1991), group 2 (1992–1996), group 3 (1997–2001), group 4 (2002–2006), group 5 (2007–2011), group 6 (2012–2016) and group 7 (2017 and later).

### Data analysis

Basic descriptive statistics were used to initially evaluate the data. The annual volume of listing and transplant of the patients stratified by race was done using area charts and evaluated using Cochran-Armitage trend test. Differences in recipient demographic and clinical factors as well as donor factors between the racial groups were evaluated. Considering the large sample size,

we limited our univariate analysis to overall racial groups and within the Black race, we performed univariate analysis to identify differences in recipient demographic and clinical factors as well as donor factors between time periods (groups 2–7). Era 1 patients were excluded from demographic and logistic regression analysis due to missing values in several of the variables especially the hemodynamic and MCS use.

The Scientific Registry of Transplant Recipients (SRTR) heart transplant risk model provides information on variables associated with 3-year mortality [12]. We created a Cox regression model (using majority of SRTR variables) and forced in the variable of transplant era as a risk factor given the improvement in the transplant care over several decades. We also assessed the interaction between the race and transplant era to evaluate racial differences in post-transplant outcomes over time. Racial differences in transplant insurance coverage have been previously identified, hence we performed the post-transplant survival analysis using Kaplan-Meier curves stratified by race, insurance type and transplant era [13]. We also evaluated discharge immunosuppression status as well as incidence of acute rejection (available after 2004) prior to discharge in these patients. Since Tacrolimus became widely used after 2005, we focused on discharge maintenance therapy of Tacrolimus and Mycophenolate, use of thymoglobulin and Basilximab as induction therapy and use of steroids as anti-rejection (prior to discharge) in patients of the last 3 eras.

Non-parametric tests (Kruskal-Wallis Test) were used to evaluate the differences between the continuous variables and chi-squared test to evaluate differences between the categorical variables. The differences in Kaplan-Meier survival curves were evaluated using the log-rank test. Data was presented in median (inter quartile range) and % (N) for univariate statistics and Hazard Ratio (HR) for multivariate statistics. The analysis was done using SAS software (SAS Inc., Cary, NC) at 95% confidence interval level.

## Results

### Listing and transplant volumes

A total of 105,266 adult patients have been listed for heart transplant since 1987, of which 67,824 have been transplanted. Of the transplanted patients, 11,235 were Black, 48,786 White and 6,803 were other race (including Hispanics). The number of patients listed gradually increased over time from 1,017 in 1987 to 4,029 in 2019. During this period, the proportion of Black patients listed for transplant increased from 7% in 1987 to 13.4% in 1999 and 25% in 2019 (Fig 1). The number of patients transplanted increased from 274 in 1987 to 3,034 in 2019. During the same period, the proportion of Black patients transplanted increased from 5% in 1987 to 13.4% in 2001 and 26% in 2019 (Fig 2).

### Demographic and risk profile

The Black patients at time of transplant were younger (51 v. 53 v. 56 years, p < .01), more likely to be female (34% v. 25% v. 22%, p < .01) and with higher BMI (27 v. 25 v. 26, p < .01) compared to White patients and other racial groups. The median creatinine (1.3 v. 1.2 v. 1.2, p < .01), and mean pulmonary artery pressure (29 v. 28 v. 27, p < .01) were higher in Black patients compared to other groups. The donor profile was clinically similar across the racial groups except Black patients were more likely to receive an organ from other racial groups (Table 1). Insurance information was recorded in the database since 1994 which showed that Black patients had a higher proportion of Medicaid insurance (20% v. 8%) and lower proportion of private insurance (42% v. 58%) compared to Whites (p < .01). Table 1 describes transplant and donor characteristics by racial groups. Within Black patients, the overall recipient clinical profile changed over time; median age and male patients increased, the pulmonary

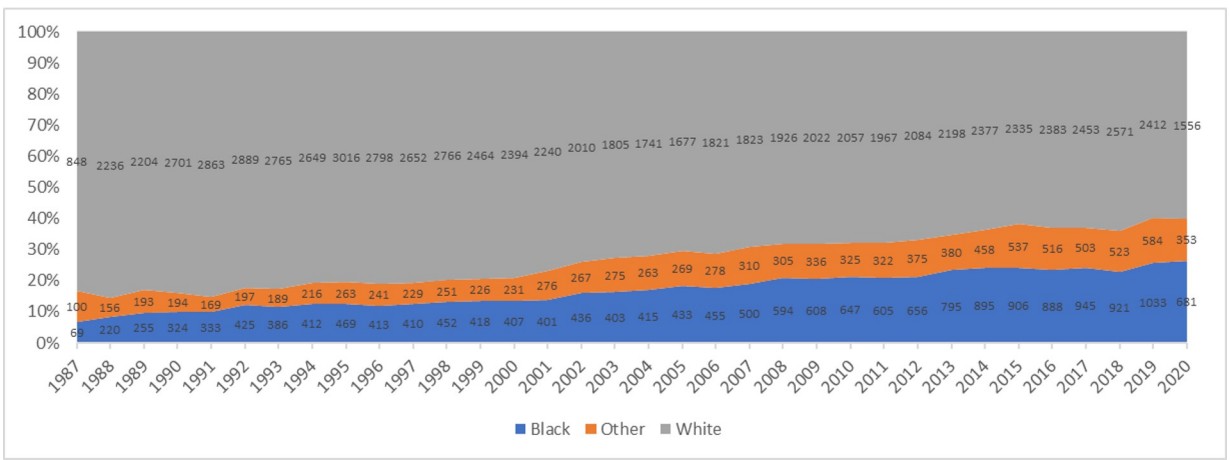

**Fig 1. Annual heart transplant listing stratified by race.**

artery pressures reduced, use of left ventricular assist devices (LVADs) increased, while the waiting time fluctuated and the donor characteristics had marginal change compared to previous era (Table 2).

## Post-transplant survival outcomes

The overall 3-year post-transplant survival improved over time for all races since 1987 (For Black patients from 70% to 83%, Whites 75% to 85% and others 72% to 84%, Fig 3). The survival gap between the Black patients and other races gradually improved with the most substantial improvement observed from 2012 to present. After stratifying the cohort by the insurance type, we observed that privately insured White patients have better 3-year survival across all groups over the years, however, the survival gap between the racial and insurance groups have reduced over time (Table 3). Black patients with private insurance had an 83% survival in the latest era (2017 onwards) compared to 85% in white patients with private insurance (p = 0.8), a substantial improvement from the 1990s when Black patients had 3-year survival <75% and White patients had over 80% (Table 3). In the 1990s, Black Medicaid patients had 3-year survival <70% which improved close to 80% in the recent era (2012–2016, Table 3)

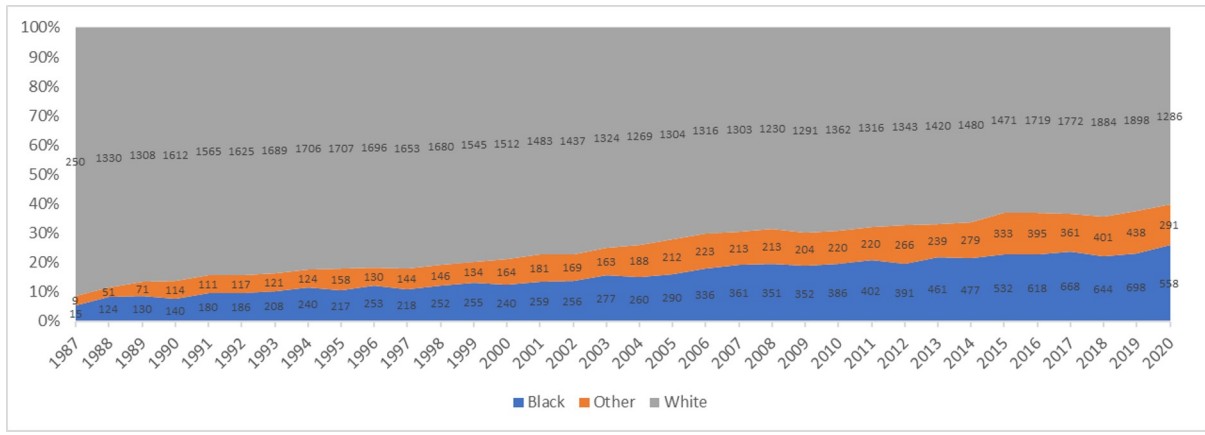

**Fig 2. Annual heart transplant volume stratified by race.**

**Table 1. Demographic and clinical risk factors stratified by race.**

| | Black (n = 10521) | Other (n = 6362) | White (n = 43001) | p-value |
|---|---|---|---|---|
| Recipient Characteristics | | | | |
| Age | 51 (41–59) | 53 (43–60) | 56 (48–62) | < .01 |
| Gender-Male | 66% | 75% | 78% | < .01 |
| BMI | 27 (23–31) | 25 (22–29) | 26 (23–30) | < .01 |
| Creatinine | 1.3 (1.0–1.6) | 1.1 (0.9–1.5) | 1.2 (1.0–1.5) | < .01 |
| Dialysis | 6% | 5% | 3% | < .01 |
| Bilirubin | 0.7 (0.5–1.2) | 0.8 (0.5–1.3) | 0.8 (0.5–1.2) | < .01 |
| Pulmonary Artery pressure (mean) | 29 (22–36) | 28 (21–36) | 27 (20–35) | < .01 |
| Insurance | | | | |
| Private | 42% | 44% | 58% | < .01 |
| Medicaid | 20% | 20% | 8% | |
| Medicare | 34% | 29% | 29% | |
| Other | 5% | 7% | 4% | |
| Ischemia Time | 3.2 (2.3–3.7) | 3.2 (2.3–3.7) | 3.2 (2.3–3.7) | 0.24 |
| Wait List duration | 83 (26–246) | 61 (19–195) | 97 (28–279) | < .01 |
| Diabetes | 26% | 30% | 23% | < .01 |
| Etiology | | | | |
| Congenital Heart Defect | 1% | 2% | 3% | < .01 |
| Ischemic Cardiomyopathy | 16% | 28% | 33% | |
| Non- Ischemic Cardiomyopathy | 62% | 46% | 36% | |
| Restrictive Cardiomyopathy | 11% | 8% | 7% | |
| Re Transplant | 2% | 3% | 3% | |
| Other | 7% | 12% | 18% | |
| Malignancy | 6% | 4% | 7% | < .01 |
| Ventilator | 1.70% | 2.20% | 2.30% | 0.02 |
| Ventricular Assist Device | 33% | 26% | 23% | < .01 |
| Total Artificial Heart | 1% | 1% | 1% | 0.10 |
| ECMO | 1% | 1% | 1% | 0.12 |
| Donor Characteristics | | | | |
| Age | 30 (22–40) | 29 (21–40) | 30 (21–41) | 0.01 |
| Gender-Male | 71% | 66% | 71% | < .01 |
| BMI | 26 (23–30) | 25 (22–29) | 26 (23–29) | < .01 |
| Diabetes | 2.60% | 2.50% | 2.50% | 0.88 |
| Race | | | | |
| Black | 19% | 12% | 13% | < .01 |
| Other | 17% | 32% | 15% | |
| White | 64% | 56% | 72% | |

ECMO = Extra-Corporeal Membrane Oxygenation, BMI = Body Mass Index, Data present Median (Inter Quartile Range) and %.

which was comparable to the 3-year survival in White Medicaid patients (81%) in the corresponding era.

## Adjusted outcomes–cox regression

The multivariate cox regression showed that the recipient BMI (HR = 1.02, p < .01), creatinine (HR = 1.02, p < .01), bilirubin (HR = 1.04, p < .01), mean PA pressures (HR = 1.01, p < .01), history of diabetes (HR = 1.08, p < .01), ischemic cardiomyopathy (HR = 1.18, p = .01),

**Table 2. Demographic and Clinical Risk Factors stratified by Era of Transplant within Black Patients.**

| Black Patients by Era | Era 2 N = 1083 | Era 3 N = 1178 | Era 4 N = 1384 | Era 5 N = 1832 | Era 6 N = 2475 | Era 7 N = 2567 | p-value |
|---|---|---|---|---|---|---|---|
| Recipient Characteristics | | | | | | | |
| Age | 48 (38–55) | 50 (40–57) | 49 (38–57) | 50 (40–58) | 53 (43–60) | 54 (44–61) | |
| Gender-Male | 65% | 63% | 66% | 66% | 67% | 69% | < .01 |
| BMI | 24 (22–28) | 26 (22–29) | 26 (23–30) | 27 (23–31) | 27 (24–32) | 28 (24–32) | |
| Creatinine | 1.3 (1.0–1.6) | 1.3 (1.0–1.6) | 1.3 (1.0–1.6) | 1.3 (1.0–1.6) | 1.3 (1.0–1.6) | 1.3 (1.0–1.6) | |
| Dialysis | 2% | 4% | 5% | 5% | 6% | 7% | < .01 |
| Bilirubin | 0.9 (0.6–1.4) | 0.8 (0.5–1.4) | 0.8 (0.5–1.5) | 0.8 (0.5–1.3) | 0.7 (0.4–1.1) | 0.7 (0.4–1.0) | |
| Pulmonary Artery pressure (mean) | 35 (26–42) | 32 (24–39) | 29 (23–36) | 30 (23–37) | 27 (21–35) | 27 (21–34) | |
| Insurance | | | | | | | |
| Private | 51% | 52% | 44% | 41% | 36% | 39% | < .01 |
| Medicaid | 22% | 19% | 18% | 22% | 18% | 19% | |
| Medicare | 20% | 24% | 30% | 35% | 40% | 37% | |
| Other | 7% | 5% | 4% | 4% | 5% | 6% | |
| Ischemia Time | 2.6 (2.0–3.3) | 3.0 (2.3–3.6) | 3.1 (2.5–3.7) | 3.1 (2.4–3.7) | 3.0 (2.3–3.7) | 3.2 (2.5–3.8) | |
| Wait List duration | 85 (30–219) | 95 (35–253) | 64 (22–179) | 75 (24–201) | 112 (35–320) | 70 (17–276) | |
| Diabetes | 12% | 19% | 22% | 28% | 29% | 31% | < .01 |
| Etiology | | | | | | | |
| CHD | 1% | 1% | 1% | 1% | 1% | 2% | < .01 |
| ICM | 5% | 20% | 20% | 19% | 17% | 13% | |
| NICM | 59% | 65% | 67% | 64% | 61% | 59% | |
| RCM | 5% | 4% | 5% | 9% | 15% | 19% | |
| ReTx | 2% | 2% | 3% | 3% | 2% | 2% | |
| Other | 28% | 8% | 5% | 4% | 4% | 4% | |
| Malignanacy | 2% | 4% | 3% | 5% | 7% | 8% | < .01 |
| Ventilator | 4% | 2% | 2% | 2% | 1% | 1% | < .01 |
| VAD | 7% | 16% | 21% | 32% | 50% | 44% | < .01 |
| TAH | 0.20% | 1% | 0.58% | 1% | 1% | 1% | |
| ECMO | 0.10% | 0.10% | 0.30% | 0.60% | 0.70% | 3.00% | < .01 |
| Doror Characteristics | | | | | | | |
| Age | 27 (18–39) | 30 (20–42) | 28 (20–40) | 30 (22–41) | 30 (22–40) | 31 (24–39) | |
| Gender-Male | 67% | 67% | 70% | 73% | 71% | 72% | < .01 |
| BMI | 24 (21–27) | 25 (22–28) | 25 (23–29) | 26 (23–30) | 27 (23–31) | 27 (24–31) | |
| Diabetes | 1% | 2% | 2% | 3% | 3% | 3% | < .01 |
| Race | | | | | | | |
| Black | 15% | 17% | 17% | 19% | 20% | 20% | < .01 |
| Other | 12% | 13% | 17% | 17% | 18% | 19% | |
| White | 73% | 70% | 66% | 64% | 61% | 61% | |

BMI = Body Mass Index, CHD = Congenital Heart Defect, ICM = Ischemic Cardiomyopathy, NICM = Non-ischemic Cardiomyopathy, RCM = Restrictive Cardiomyopathy, ReTx = Re Transplant, VAD = Ventricular Assist Device, TAH = Total Artificial Heart, ECMO = Extra-Corporeal Membrane Oxygenation, Data present Median (Inter Quartile Range) and %.

congenital heart defects (HR = 1.86, p < .01), increased ischemia time (HR = 1.12, p < .01), requirement of ventilator (HR = 1.87, p < .01 (1.58–2.37)) or ECMO (HR = 1.73, p < .01) were associated with increased risk of 3-year post-transplant mortality while the male recipient gender (HR = 0.91, p = 0.01) was a protective factor (Table 4 describes full cox regression model). Black patients had significantly higher risk of 3-year mortality in the ERA 4 and 5,

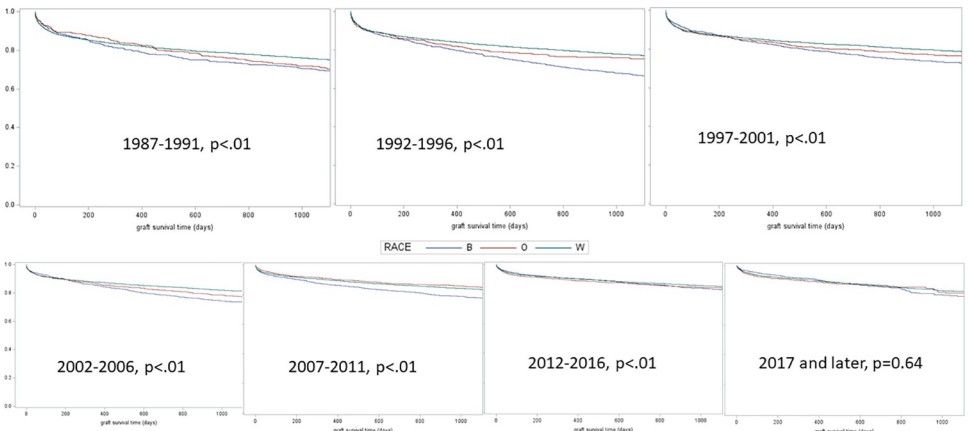

**Fig 3. Three-year post-transplant survival stratified by race and era of transplant.**

however, the race was no longer a significantly associated factor in the most recent eras (HR (ERA 6 B v W) = 1.15, p = 0.15, HR (ERA 7 B v W) = 1.00, p = 0.14). Table 4 shows details of the cox regression model.

## Immunosuppression and rejection

The use of tacrolimus and mycophenolate as maintenance therapy gradually increased since their availability and has been over 90% since 2012 for all racial groups (Fig 4). The use of steroids for anti-rejection prior to discharge in eras 5, 6 and 7 was 7%, 9% and 9% respectively for Whites, 9%, 11%, and 10% for Blacks and 8%, 9% and 9% for Other races. The number of patients treated for acute rejection remained significantly high in Black patients in era 5 (11% v. 8%, v. 10%, p < .01), 6 (13% v. 10% v. 10%, p < .01), and 7 (12% v. 10% v. 10%, p = 0.1) compared to Whites and Other races respectively. At 3 years post-transplant cardiac arrest remained the most common primary and contributing cause of death for Black patients which was significantly higher compared to Whites and Other races (era 5–4% v. 2.5% v. 1.7%,

**Table 3. Three-year post-transplant survival stratified by race and type of insurance.**

| 3-Year Post Transplant survival (%) | 1987–1991 | 1992–1996 | 1997–2001 | 2002–2006 | 2007–2011 | 2012–2016 | 2016–2020 |
|---|---|---|---|---|---|---|---|
| Private Insurance | | | | | | | |
| White | 45 | 79 | 80 | 83 | 84 | 86 | 85 |
| Black | | 73 | 74 | 77 | 78 | 84 | 83 |
| Other | | 82 | 76 | 76 | 89 | 85 | 83 |
| p-value | | < .01 | < .01 | < .01 | < .01 | 0.12 | 0.38 |
| Medicare Insurance | | | | | | | |
| White | 58 | 77 | 77 | 78 | 83 | 83 | 79 |
| Black | 66 | 74 | 75 | 73 | 80 | 82 | 79 |
| Other | | 71 | 76 | 78 | 79 | 83 | 80 |
| p-value | | 0.66 | 0.02 | 0.02 | 0.44 | 0.76 | 0.82 |
| Medicaid Insurance | | | | | | | |
| White | 75 | 73 | 74 | 79 | 83 | 81 | 82 |
| Black | | 56 | 68 | 69 | 74 | 81 | 73 |
| Other | | 82 | 71 | 77 | 83 | 84 | 81 |
| p-value | | < .01 | < .01 | < .01 | < .01 | 0.01 | 0.43 |

**Table 4. Cox regression model for post-transplant mortality at 3-years.**

| Variables | Hazard Ratio | p-value |
|---|---|---|
| **Recipient Factors** | | |
| Recipient Age (years) | 1.00 | 0.15 |
| Recipient Body Mass Index | 1.02 | < .01 |
| RECIPIENT GENDER Male v Female | 0.91 | 0.01 |
| Serum Creatinine at Time of Transplant | 1.02 | 0.01 |
| Total Bilirubin at Transplant | 1.04 | < .01 |
| Mean Pulmonary Artery pressure | 1.01 | < .01 |
| RACE Black v. White | 1.00 | 0.97 |
| RACE Other v. White | 1.00 | 0.99 |
| ERA 2 v. 7 | 2.07 | < .01 |
| ERA 3 v. 7 | 1.65 | < .01 |
| ERA 4 v. 7 | 1.21 | 0.00 |
| ERA 5 v. 7 | 1.01 | 0.90 |
| ERA 6 v. 7 | 0.90 | 0.05 |
| RACE Black vs Oth At ERA = 2 | 1.535 | 0.59 |
| RACE Black vs White At ERA = 2 | 0.959 | 0.89 |
| RACE Oth vs White At ERA = 2 | 0.625 | 0.43 |
| RACE Black vs Oth At ERA = 3 | 0.69 | 0.08 |
| RACE Black vs White At ERA = 3 | 1.183 | 0.38 |
| RACE Oth vs White At ERA = 3 | 1.716 | 0.02 |
| RACE Black vs Oth At ERA = 4 | 1.127 | 0.75 |
| RACE Black vs White At ERA = 4 | 1.575 | < .01 |
| RACE Oth vs White At ERA = 4 | 1.397 | 0.01 |
| RACE Black vs Oth At ERA = 5 | 1.523 | < .01 |
| RACE Black vs White At ERA = 5 | 1.381 | < .01 |
| RACE Oth vs White At ERA = 5 | 0.907 | 0.45 |
| RACE Black vs Oth At ERA = 6 | 1.081 | 0.57 |
| RACE Black vs White At ERA = 6 | 1.152 | 0.14 |
| RACE Oth vs White At ERA = 6 | 1.066 | 0.60 |
| RACE Black vs Oth At ERA = 7 | 1.002 | 0.59 |
| RACE Black vs White At ERA = 7 | 1.003 | 0.14 |
| RACE Oth vs White At ERA = 7 | 1.001 | 0.57 |
| Insurance Medicaid v. Private | 1.27 | < .01 |
| Insurance Medicare v. Private | 1.23 | < .01 |
| Insurance Oth v. Private | 1.03 | 0.73 |
| Etiology Arrhythomogenic cardiomyopathy v. Oth | 0.82 | 0.64 |
| Etiology Congenital Heart Disease v. Oth | 1.86 | < .01 |
| Etiology Hypertrophic Cardiomyopathy v. Oth | 0.93 | 0.53 |
| Etiology Ischemic Cardiomyopathy v. Oth | 1.18 | 0.01 |
| Etiology Non-ischemic Cardiomyopathy v. Oth | 0.97 | 0.67 |
| Etiology Restrictive Cardiomyopathy v. Oth | 1.03 | 0.66 |
| Etiology Re-transplant v. Oth | 1.40 | 0.00 |
| Diabetes 1 v. 0 | 1.08 | 0.01 |
| Extra Corporeal Membrane Oxygenation 1 v 0 | 1.73 | < .01 |
| Ventilator 1 v. 0 | 1.87 | < .01 |
| **Mechanical Circulatory Support** | | |
| BiV v. None | 1.80 | < .01 |

(*Continued*)

**Table 4.** (Continued)

| Variables | Hazard Ratio | p-value |
|---|---|---|
| CFVAD v. None | 1.20 | < .01 |
| LVAD v. None | 1.11 | 0.04 |
| TAH v. None | 1.95 | < .01 |
| Dialysis 1 v. 0 | 1.63 | < .01 |
| Previous Malignancy 1 v. 0 | 1.17 | < .01 |
| **Donor Factors** | | |
| DONOR AGE (YRS) | 1.01 | < .01 |
| DONOR GENDER Male v. Female | 0.93 | 0.03 |
| Donor Body Mass Index | 0.99 | < .01 |
| Donor Diabetes 1 v. 0 | 1.11 | 0.17 |
| Ischemic Time (hours) | 1.12 | < .01 |

ERA2 = 1992–96, ERA3 = 1997–01, ERA4 = 2002–06, ERA5 = 2007–11, ERA6 = 2012–2016, ERA7 = 2017onwards.

BiV = Bivetricular assist device, CFVAD = Continuous Flow ventricular assist device, LVAD = Pulsatile Flow ventricular assist device, TAH = Total Artificial Heart.

p<0.1; era 6–3% v. 1.5% v. 1.3%, p < .01). Black patients had significantly higher rates of chronic rejection-related mortality at 3-year as well (Table 5).

## Discussion

Heart failure is one of the most significant health burdens of our time and so are the racial disparities [8, 14]. According to the CDC reports, Black patients have significantly higher rates of heart disease-related mortality in almost all of the United States [15]. The greater burden of heart disease in the Black community also transpires to increased refractory heart failure requiring a heart transplant [15, 16]. Based on our analysis we found that in the late 80s and much of the 90s the proportion of Black patients listed and transplanted for heart were below the national proportion of the Black patients (13.4%) [17]. However, heart transplant listing and transplantation in the Black patients gradually improved over time (from 10% in 1990 to 15% in 2000), especially since the early 2000s. Since 2005, the proportion of Black patients listed and transplanted has remained over 15%. The proportion of Black patients transplanted breached the 20% mark in 2011 and the 25% mark in 2018. Our study also showed that along with improvement in listing and transplant rates, the post-transplant survival in the recent years also improved for the black patients.

We observed that the risk profile of the Black patient also changed over the years as the more recent patients are older, more likely male, on dialysis and supported by a LVAD with reduced pre-transplant PA pressures. Overall, the Black patients were younger than White patients and more likely male which is consistent with the literature showing male Black patients are at higher risk of cardiac disease at a younger age [16, 18]. At the same time, donor risk profile of Black recipients only marginally changed over the years with increase of median age from 29 to 31 years, increase in BMI from 25 to 27 and fewer White donors transplanted to Black recipients. We also observed lower median waitlist duration for the Black patients which could be due to delayed listing and higher acuity of patients as we observed that 27% of Black patients were transplanted as Status 1A compared to 21% of White patients. In a recent publication Chouairi et al. corroborated our findings of improved listing and transplant rates amongst Black patients however, they identified Black race as risk factor for poor post-transplant survival [19]. We evaluated the Race along with different transplant eras thus we

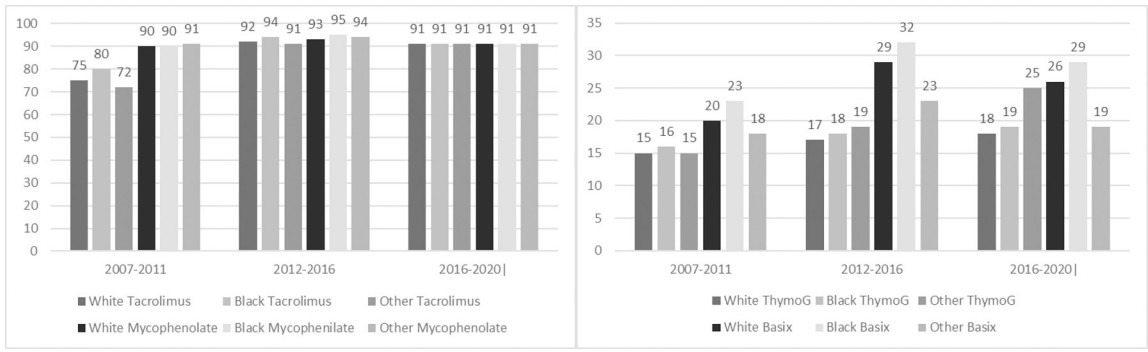

**Fig 4.** Discharge Immunosuppression by Race and Era; A) Maintenance Therapy, B) Induction Therapy.

observed that the Black patients, historically had poor survival however, in recent years the survival has improved which evident from the Kaplan-Meier curves as well as the risk-adjusted Cox regression analysis.

Racial disparities in other solid organ transplantations have been reported in terms of access as well as outcomes however few studies have done chronological comparisons [20–22]. These studies show that racial disparities have reduced in terms of kidney transplant listing in recent years following a policy change however for lung and liver transplant patients non-Hispanic Black patients continue to have poor access and outcomes [20–23]. We also observed that number of Black donors to Black recipients has also increased gradually and more so in recent years however geographical limitations in donor heart allocation may not allow a full racial matching. Although we did not assess poor survival with racial mismatch has been previously observed [24].

Black patients not only had a gradual increase in access to transplantation, but also had improved post-transplant survival at 3-years in the most recent era. Very similar to the listing and transplant statistics, the survival of Black patients had been significantly low since 1987 and remained such throughout the 1990s and 2000s. Since 2012, the survival trends for Black patients have gradually improved and have more or less been comparable to rest of the patients. Other studies have corroborated our findings of worse survival trends in Black patients however, our findings of improved and comparable survival in the recent years have not been reported [9, 10]. Although we did not assess the center level variations, we believe that improved survival in Black patients might be multi-factorial. General improvement in care, better compliance with the immunosuppression, reliable insurance coverage and increased awareness about reducing racial disparities in recent years could be contributing factors in improved survival of Black patients [25]. The adjusted outcomes using the regression

**Table 5. Cause of death in era 5 and 6.**

| Cause of Death | 2007–2011 | p-value era5* | 2012–2016 | p-value era6* |
|---|---|---|---|---|
| Black Chronic Rejection | 1.50% | < .01 | 1% | < .01 |
| White Chronic Rejection | 0.40% | | 0.30% | |
| Other Chronic Rejection | 0.70% | | 1% | |
| Black Cardiac Arrest | 4% | < .01 | 3% | < .01 |
| White Cardiac Arrest | 1.70% | | 1.30% | |
| Other Cardiac Arrest | 2.70% | | 1.50% | |

*p-value between the racial groups within the era.

model (like the SRTR) showed an interaction between race and the era of transplant, suggestive of gradual improvement in survival of Black patients. Further, Black race was not a significant predictor of mortality post-transplant in the most recent era, indicating the improvement in outcomes for Black patients in recent years. The annual registry data also corroborates our finding of improved survival of transplant patients overtime [26].

The Medicare and Medicaid patients had poorer post-transplant survival, however the survival of these patients improved over time but remained marginally below the private insurance patients. The improvement in survival of Black patients was seen through all insurance classes over time particularly in the last 2 eras, which could be partially due to better insurance coverage after the application of the Affordable Care Act [27, 28].

Cardiac arrest remained the most common primary cause of death post-transplant in Black patients, however it marginally trended downwards in the most recent era. We did not identify a significant difference in discharge immunosuppression in the Black patients, however use of Simulect as induction therapy was significantly higher in Black patients compared to the other groups, especially after 2012. We did not observe significant improvement in survival with use of newer agents [26]. Cardiac allograft vasculopathy has been recognized as the leading cause of graft failure overall which could be more pronounced in Black patients as more patients have died of cardiac arrest and chronic rejection than other racial groups.

## Limitations

This is a retrospective database-driven study and relied on the previously collected data over a span of 33 years. An important weakness was that all the data points were not available or collected since the beginning, particularly the insurance information which was not collected prior to 1994. Similarly, the information on acute rejection was only available after 2004. Although we used insurance as an important surrogate of socio-economic status, we could not perform an analysis to evaluate its impact if there was a change in the insurance type during the course of the follow-up and its impact on the post-transplant survival.

## Conclusion

This structurally collected serial cardiac transplant data since 1987 shows that listings and transplants remained below 12% for Black patients during the 1980s and 1990s even though the rates of heart failure were higher than other races. Access to transplantation broke even to the national proportion of Black population (13.4%) in 2001 when 13.4% of all adult cardiac transplants performed were in Black patients. Since 2001 more Black patients have been listed and transplanted breaching the 20% mark in 2010 and 25% mark in 2019. Similar to the improved access, the survival of Black patients have improved more recently (since 2012) when their historically poor outcomes improved and became comparable to rest of the patients.

## Acknowledgments

This work was supported in part by Health Resources and Services Administration contract 234-2005-37011C. The content is the responsibility of the authors alone and does not necessarily reflect the views or policies of the Department of Health and Human Services nor does mention of trade names, commercial products, or organizations imply endorsement by the US government.

## Author Contributions

**Conceptualization:** Jaimin R. Trivedi, Mark S. Slaughter.

**Investigation:** Jaimin R. Trivedi, Siddharth V. Pahwa.

**Methodology:** Jaimin R. Trivedi, Mark S. Slaughter.

**Project administration:** Jaimin R. Trivedi.

**Supervision:** Jaimin R. Trivedi.

**Writing – original draft:** Jaimin R. Trivedi, Katherine R. Whitehouse.

**Writing – review & editing:** Siddharth V. Pahwa, Bradley M. Ceremuga, Mark S. Slaughter.

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
