## [Decision Letter · Decision Letter 0]

5 Oct 2021

PONE-D-21-27571Racial Disparities in Cardiac Transplantation: Chronological Perspective and OutcomesPLOS ONE

Dear Dr. Trivedi,

Thank you for submitting your manuscript to PLOS ONE. After careful consideration, we feel that it has merit but does not fully meet PLOS ONE’s publication criteria as it currently stands. Therefore, we invite you to submit a revised version of the manuscript that addresses the points raised during the review process. Please ensure that the definitions on race and ethnicity are in line with the instruction to authors.

We look forward to receiving your revised manuscript.

Kind regards,

Antonio Cannatà

Academic Editor

PLOS ONE

Journal Requirements:

 [No]. 

[No authors have competing interests.]

Additional Editor Comments:

Please resentence the manuscript to avoid misinterpretation of racial disparities and to adhere to scientific standards.

Reviewers' comments:

Reviewer's Responses to Questions

**Comments to the Author**

1. Is the manuscript technically sound, and do the data support the conclusions?

Reviewer #1: Yes

Reviewer #2: Partly

2. Has the statistical analysis been performed appropriately and rigorously? 

Reviewer #1: Yes

Reviewer #2: No

3. Have the authors made all data underlying the findings in their manuscript fully available?

Reviewer #1: Yes

Reviewer #2: No

4. Is the manuscript presented in an intelligible fashion and written in standard English?

Reviewer #1: Yes

Reviewer #2: Yes

5. Review Comments to the Author

Reviewer #1: In this paper, authors aimed to evaluate annual heart transplant volumes and 3-year post-transplant outcomes of patients in UNOS database stratifying them by race. The topic is interesting and deserves space, however a very recent article has been already published from UNOS database: Chouairi F. et al. Evaluation of Racial and Ethnic Disparities in Cardiac Transplantation. J Am Heart Assoc. 2021 Sep 7;10(17):e021067.

For this reason, I would suggest:

1) to add some comments on this recent paper, where the data about the increased number in transplant listing of Black patients are similar, but the data on mortality seems different, as far as Chouairi showed a higher risk of post‐transplantation death for Black patients, not confirmed here.

2) it would be interesting to compare this data with those ones on other solid organs, such as lung transplant patients too

(Riley LE, Lascano J. Gender and racial disparities in lung transplantation in the United States. J Heart Lung Transplant. 2021 Sep;40(9):963-969. doi: 10.1016/j.healun.2021.06.004. Epub 2021 Jun 12).

3) how do the authors explain that Black patients were transplanted less but their time in the waiting list is shorter than white patients?

4) I think that it is not very appropriate the use of some sentences in the Discussion, such as "Black lives did not matter" and “Black lives have mattered more now”, I would suggest to re-phrase these sentences without this provocative tone;

5) In the Introduction, where the authors briefly summarized the novelties on heart transplantation during the different eras (cyclosporine, MCS, ex vivo perfusion...), it would be interesting to cite also the fact that the recipients selection is expanding over the years, whereas some diagnosis that were considered contraindicated for heart transplants some years ago have reached the candidability in the recent time (see for example Di Nora C. Heart transplantation in cardiac storage diseases: data on Fabry disease and cardiac amyloidosis. Curr Opin Organ Transplant. 2020 Jun;25(3):211-217).

6) did the authors test in the multivariate logistic regression also these variables: mismatch D/R? pulmonary artery pressure? IABP or other MCS support befor heart transplantation?

7) Black recipients are less than White ones across the eras, but also Black donors are less compared to Black ones. I think that this concept deserves some comments on the Discussion (Is it due to the allocation system?)

Minor points:

- correct the spelling in Table 5

- In the Introduction, authors cited ex-vivo perfusion, however, they did not use any references on ex-vivo perfusion, I suggest to add almost one about it, see for example: Sponga S et al Heart transplant outcomes in patients with mechanical circulatory support: cold storage versus normothermic perfusion organ preservation. Interact Cardiovasc Thorac Surg. 2021 Apr 8;32(3):476-482.

Reviewer #2: The Authors of the manuscript, entitled "Racial Disparities in Cardiac Transplantation: Chronological Perspective and Outcomes", present data from United Network for Organ Sharing (UNOS) database stratified by race with focus on changes in the course of the last 30 years. They analyzed data of 105266 adults listed for heart transplantation and 67824 who have been transplanted. The important message is the improvement in the accessibility and survival after heart translantation of Black patients. The advantage of the manuscript is the clear message based on the high number of cases coming from long term registry. The registry covered demographic and clinical data. The manuscript can be valuable for PLOS ONE but the quality of this work requires improvement.

Comments:

1. Introduction:

- The abbreviation UNOS was not explained in the text, only was mentioned in the abstract.

2. Methods:

- Please report the number of centers and characterise the registry.

- Please describe the method of data acquisition, data verification, method of data quality check, missing values imputation, if done?

- The number of missing data in relation to the presented variables should be presented in the main tables or as the supplement.

- What was the rationale for using logistic regression? For mortality analysis it is more reasonable to use the cox regression model.

3. Results:

- Please consistently present p-values where differences between groups are presented.

- Tables: Some of the variables are in abbreviation but the abbreviations are not explained, for example BMI, PA, CHD, etc only ECMO is explained. The Authors did not payed enough attention to the quality of data presentation. The units are missing.  

- In the section post-transplant survival outcomes Authors describe the changes in the mortality. We can observe numerical differences and suggestive changes in the course of different periods. Testing for difference was performed only in one place.  

- In the Table 3 - Three-year post-transplant survival stratified by race and type of insurance there is not shown any p-value. I would suggest adding p-value for testing between races in a specific subgroups.

- The BMI, ECMO abbreviation is not explained inthe text.

- Table 4: It is not clear if presented odds ratios with 95% limits are for univariate or multivariate analysis? I would suggest showing both univariate and multivariate odds ratios with 95% limits in the table 4 (like mentioned above optimally the cox regression model). Variables that were statistically significant in univariate tests should be used for multivariate testing.

- In the immunosuppression and rejection section is the lack of p-value in the sentence describing post-transplant cardiac arrest. 

5. Discussion:

- In the first sentences of the discussion I would suggest summarizing the key findings of the paper.

- Sentence:”Based on our analysis we found that “Black lives did not matter” in the late 80s and much of the 90s when the proportion of Black patients listed for and transplanted were below the national proportion of the Black patients (13.4%)”. Please include the percentage of listed patients and national proportion of Black people.

- Please comment also and include adequate references about the incidence of heart failure in relation to race and confront with study results.

- Why Black patients were younger, more likely to be female and with higher BMI compared to White patients and other racial groups? The differences should be discussed.

6. PLOS authors have the option to publish the peer review history of their article (what does this mean?). If published, this will include your full peer review and any attached files.

Reviewer #1: No

Reviewer #2: **Yes: **Mateusz Sokolski

---

## [Author Response · Author response to Decision Letter 0]

15 Nov 2021

Reviewer #1: In this paper, authors aimed to evaluate annual heart transplant volumes and 3-year post-transplant outcomes of patients in UNOS database stratifying them by race. The topic is interesting and deserves space, however a very recent article has been already published from UNOS database: Chouairi F. et al. Evaluation of Racial and Ethnic Disparities in Cardiac Transplantation. J Am Heart Assoc. 2021 Sep 7;10(17):e021067.

We thank reviewer for his/her comments and suggestions, including the JAHA article which was published after we submitted our work. Below is our pointwise response to all the comments and suggestions. 

For this reason, I would suggest:

1) to add some comments on this recent paper, where the data about the increased number in transplant listing of Black patients are similar, but the data on mortality seems different, as far as Chouairi showed a higher risk of post‐transplantation death for Black patients, not confirmed here.

A1) We have included the aforementioned article in the references as well added text in the discussion section about the article and differences in findings.

2) it would be interesting to compare this data with those ones on other solid organs, such as lung transplant patients too

(Riley LE, Lascano J. Gender and racial disparities in lung transplantation in the United States. J Heart Lung Transplant. 2021 Sep;40(9):963-969. doi: 10.1016/j.healun.2021.06.004. Epub 2021 Jun 12).

A2) We have added the comparison of our findings to other solid organ transplants in the discussion section.

3) how do the authors explain that Black patients were transplanted less but their time in the waiting list is shorter than white patients?

A3) We believe that Black patients are listed late and possibly with higher acuity and priority status as evidenced by greater presence of risk factors. This could be responsible for the shorter waitlist duration. We observed that 27% of Black patients were transplanted as Status 1A compared to 21% of White patients. We have added this in the discussion section.

4) I think that it is not very appropriate the use of some sentences in the Discussion, such as "Black lives did not matter" and “Black lives have mattered more now”, I would suggest to re-phrase these sentences without this provocative tone;

A4) We have removed the controversial texts and replaced it with more appropriate and scientific terminology.

5) In the Introduction, where the authors briefly summarized the novelties on heart transplantation during the different eras (cyclosporine, MCS, ex vivo perfusion...), it would be interesting to cite also the fact that the recipients selection is expanding over the years, whereas some diagnosis that were considered contraindicated for heart transplants some years ago have reached the candidability in the recent time (see for example Di Nora C. Heart transplantation in cardiac storage diseases: data on Fabry disease and cardiac amyloidosis. Curr Opin Organ Transplant. 2020 Jun;25(3):211-217).

A5) We have added the aforementioned citation.

6) did the authors test in the multivariate logistic regression also these variables: mismatch D/R? pulmonary artery pressure? IABP or other MCS support befor heart transplantation?

A6) We have evaluated PA pressures, MCS device use including ECMO as part of the multivariate regression model. We have also included recipient and donor race as part of the regression model independently to show their impact on survival. However, we have not tested the D/R race mismatch as a factor as we were focusing on Recipient race and era of transplantation. 

7) Black recipients are less than White ones across the eras, but also Black donors are less compared to Black ones. I think that this concept deserves some comments on the Discussion (Is it due to the allocation system?)

A7) The Black recipients as well as donors are less than White across eras however, this is possibly due to general racial distribution of population where there are fewer Black people than White thus it will be unlikely that there would be more Black recipients or donors than White patients. Although, we have observed that racial mismatch is donation is decreasing and a higher proportion of Black donor hearts have been used for transplant. We have added this to the discussion section. 

Minor points:

- correct the spelling in Table 5

- In the Introduction, authors cited ex-vivo perfusion, however, they did not use any references on ex-vivo perfusion, I suggest to add almost one about it, see for example: Sponga S et al Heart transplant outcomes in patients with mechanical circulatory support: cold storage versus normothermic perfusion organ preservation. Interact Cardiovasc Thorac Surg. 2021 Apr 8;32(3):476-482.

We have addressed these minor points.

Reviewer #2: The Authors of the manuscript, entitled "Racial Disparities in Cardiac Transplantation: Chronological Perspective and Outcomes", present data from United Network for Organ Sharing (UNOS) database stratified by race with focus on changes in the course of the last 30 years. They analyzed data of 105266 adults listed for heart transplantation and 67824 who have been transplanted. The important message is the improvement in the accessibility and survival after heart translantation of Black patients. The advantage of the manuscript is the clear message based on the high number of cases coming from long term registry. The registry covered demographic and clinical data. The manuscript can be valuable for PLOS ONE but the quality of this work requires improvement.

We thank the reviewer for the insightful comments and suggestions. We have replace the logistic regression with the Cox regression analysis. Your overall suggestions have truly improved the quality of this manuscript.

Comments:

1. Introduction:

- The abbreviation UNOS was not explained in the text, only was mentioned in the abstract.

A1) This abbreviation is added.

2. Methods:

- Please report the number of centers and characterise the registry.

- Please describe the method of data acquisition, data verification, method of data quality check, missing values imputation, if done?

- The number of missing data in relation to the presented variables should be presented in the main tables or as the supplement.

- What was the rationale for using logistic regression? For mortality analysis it is more reasonable to use the cox regression model.

A2) 

-Number of centers have been added during each era. 

-Data management related sentences have been added in the methods sections.

-We have added sentences about missing data values in the methods section.

-We have performed the cox-regression analysis on the same data as well and have replaced the logistic regression with the cox regression.

3. Results:

- Please consistently present p-values where differences between groups are presented.

- Tables: Some of the variables are in abbreviation but the abbreviations are not explained, for example BMI, PA, CHD, etc only ECMO is explained. The Authors did not payed enough attention to the quality of data presentation. The units are missing. 

- In the section post-transplant survival outcomes Authors describe the changes in the mortality. We can observe numerical differences and suggestive changes in the course of different periods. Testing for difference was performed only in one place. 

- In the Table 3 - Three-year post-transplant survival stratified by race and type of insurance there is not shown any p-value. I would suggest adding p-value for testing between races in a specific subgroups.

- The BMI, ECMO abbreviation is not explained inthe text.

- Table 4: It is not clear if presented odds ratios with 95% limits are for univariate or multivariate analysis? I would suggest showing both univariate and multivariate odds ratios with 95% limits in the table 4 (like mentioned above optimally the cox regression model). Variables that were statistically significant in univariate tests should be used for multivariate testing.

- In the immunosuppression and rejection section is the lack of p-value in the sentence describing post-transplant cardiac arrest. 

A3)

-We have streamlined the p-values.

- We have updated the tables to fully explain the text

-We have added the p-values in table 3 that describe insurance related differences in survival overtime.

-We have added p-values in table 3.

-We have added the necessary abbreviations

-We have replaced the logistic regression model with cox regression model.

 - p-value is added for cardiac arrest.

5. Discussion:

- In the first sentences of the discussion I would suggest summarizing the key findings of the paper.

- Sentence:”Based on our analysis we found that “Black lives did not matter” in the late 80s and much of the 90s when the proportion of Black patients listed for and transplanted were below the national proportion of the Black patients (13.4%)”. Please include the percentage of listed patients and national proportion of Black people.

- Please comment also and include adequate references about the incidence of heart failure in relation to race and confront with study results.

- Why Black patients were younger, more likely to be female and with higher BMI compared to White patients and other racial groups? The differences should be discussed.

A5)

-The key findings have been summarized

-We removed the controversial text and added technically appropriate text.

-We added the appropriate references for heart failure prevalence

-We have expanded the discussion on young Black male patients.

---

## [Decision Letter · Decision Letter 1]

10 Jan 2022

Racial Disparities in Cardiac Transplantation: Chronological Perspective and Outcomes

PONE-D-21-27571R1

Dear Dr. Trivedi,

We’re pleased to inform you that your manuscript has been judged scientifically suitable for publication and will be formally accepted for publication once it meets all outstanding technical requirements.

Kind regards,

Antonio Cannatà

Academic Editor

PLOS ONE

Additional Editor Comments (optional):

Reviewers' comments:

Reviewer's Responses to Questions

**Comments to the Author**

1. If the authors have adequately addressed your comments raised in a previous round of review and you feel that this manuscript is now acceptable for publication, you may indicate that here to bypass the “Comments to the Author” section, enter your conflict of interest statement in the “Confidential to Editor” section, and submit your "Accept" recommendation.

Reviewer #1: All comments have been addressed

Reviewer #2: All comments have been addressed

2. Is the manuscript technically sound, and do the data support the conclusions?

Reviewer #1: (No Response)

Reviewer #2: Yes

3. Has the statistical analysis been performed appropriately and rigorously? 

Reviewer #1: (No Response)

Reviewer #2: Yes

4. Have the authors made all data underlying the findings in their manuscript fully available?

Reviewer #1: (No Response)

Reviewer #2: Yes

5. Is the manuscript presented in an intelligible fashion and written in standard English?

Reviewer #1: (No Response)

Reviewer #2: Yes

6. Review Comments to the Author

Reviewer #1: (No Response)

Reviewer #2: The authors provided a revised version of their analysis and addressed all comments. I recommend acceptance of the manuscript.

7. PLOS authors have the option to publish the peer review history of their article (what does this mean?). If published, this will include your full peer review and any attached files.

Reviewer #1: No

Reviewer #2: No

---

## [Editor Report · Acceptance letter]

17 Jan 2022

PONE-D-21-27571R1 

Racial Disparities in Cardiac Transplantation: Chronological Perspective and Outcomes 

Dear Dr. Trivedi:

I'm pleased to inform you that your manuscript has been deemed suitable for publication in PLOS ONE. Congratulations! Your manuscript is now with our production department. 

Kind regards, 

on behalf of

Dr. Antonio Cannatà 

Academic Editor

PLOS ONE